

# EMT-related gene expression is positively correlated with immunity and may be derived from stromal cells in osteosarcoma

Yin-xiao Peng[1],[*], Bin Yu[1],[*], Hui Qin[1], Li Xue[1], Yi-jian Liang[1] and Zheng-xue Quan[2]

[1] Department of Orthopedics, The Third People's Hospital of Chengdu, Chengdu, Sichuan, China
[2] Department of Orthopedics, The First Affiliated Hospital of Chongqing Medical University, Chongqing, Chongqing, China
[*] These authors contributed equally to this work.

## ABSTRACT

**Background:** Osteosarcoma is the most common type of bone cancer in children and young adults. Recent studies have shown a correlation between epithelial–mesenchymal transition (EMT)-related gene expression and immunity in human cancers. Here, we investigated the relationship among EMT, immune activity, stromal activity and tumor purity in osteosarcoma.

**Methods:** We defined EMT gene signatures and evaluated immune activity and stromal activity based on the gene expression and clinical data from three independent microarray datasets. These factors were evaluated by single sample Gene Set Enrichment Analyses and the ESTIMATE tool. Finally, we analyzed the key source of EMT gene expression in osteosarcoma using microarray datasets from the Gene Expression Omnibus and human samples that we collected.

**Results:** EMT-related gene expression was positively correlated with immune and stromal activity in osteosarcoma. Tumor purity was negatively correlated with EMT, immune activity and stromal cells. We further demonstrated that high EMT gene expression could significantly predict poor overall survival (OS) and recurrence-free survival (RFS) in osteosarcoma patients, while high immune activity cannot. However, combining these factors could have further prognostic value for osteosarcoma patients in terms of OS and RFS. Finally, we found that stromal cells may serve as a key source of EMT gene expression in osteosarcoma.

**Conclusion:** The results of this study reveal that the expression of EMT genes and immunity are positively correlated, but these signatures convey disparate prognostic information. Furthermore, the results indicate that EMT-related gene expression may be derived from stromal rather than epithelial cancer cells.

# INTRODUCTION

Osteosarcoma is the most common type of bone cancer in children and young adults and is responsible for approximately 9% of cancer deaths in children and adolescents aged 10–24 years (*Biazzo & De Paolis, 2016*; *Maximov et al., 2019*). Chemotherapy and surgical intervention can increase the 5 year survival rate to 60–70%. However, the 5 year survival

Corresponding authors
Bin Yu, 768832674@qq.com
Zheng-xue Quan,
quanzx18@126.com

rate is below 30% for patients who have metastatic disease at diagnosis or recurrence (*Sayles et al., 2019*; *Shen et al., 2017*). Thus, it is important to explore the mechanisms underlying osteosarcoma metastasis to improve patient prognosis.

Tumor metastasis begins with epithelial–mesenchymal transition (EMT), in which epithelial cells acquire a mesenchymal phenotype and become migratory and invasive (*Shen et al., 2017*; *Wang et al., 2018*). Loss of the epithelial marker E-cadherin and an increase in the expression of mesenchymal markers are characteristics of EMT. EMT promotes primary epithelial-like tumor cells to acquire invasive mesenchymal phenotypes during metastatic progression. This progression, along with increased motility and invasiveness, triggers the dissemination of metastatic cells from the tumor, which then infiltrate into the tumor vasculature (*Lv et al., 2016*). Hence, EMT-related genes are potential markers and therapeutic targets for osteosarcoma treatment.

A recent study showed that EMT is associated with immunity in human cancers (*Gugnoni et al., 2017*; *Mitschke, Burk & Reinheckel, 2019*; *Yeung & Yang, 2017*). Increased expression of EMT markers in breast tumors is associated with increased immune infiltration into the tumor microenvironment (*Kotiyal & Bhattacharya, 2014*; *Yeung & Yang, 2017*). The presence of these immune cells then promotes immune evasion by the tumor cells, which is associated with tumor progression and metastasis (*Singh & Chakrabarti, 2019*). EMT status was found to be strongly associated with an inflammatory tumor microenvironment in non-small-cell lung cancer (*Lou et al., 2016*). Pancancer analysis revealed a strong correlation between EMT and immune activation (*Mak et al., 2016*). However, to the best of our knowledge, the relationship between EMT and the immune microenvironment in osteosarcoma has not been reported. What is the cellular origin of EMT-related gene signatures in osteosarcoma expression data? How do EMT genes and immune activity collectively impact survival in osteosarcoma patients?

In this study, we explored the relationship among EMT, immune activity, stromal activity and tumor purity in osteosarcoma based on gene expression and clinical data from three independent microarray datasets. These factors were evaluated by single sample Gene Set Enrichment Analysis (ssGSEA) and the ESTIMATE tool. Finally, real-time quantitative polymerase chain reaction (RT-qPCR) was used to verify the results.

## MATERIALS AND METHODS

### Raw data

Three independent microarray datasets, namely, GSE36001, GSE39055 and GSE16091, were obtained from the Gene Expression Omnibus database (GEO, https://www.ncbi.nlm. nih.gov/geo/) (*Kelly et al., 2013*; *Kresse et al., 2012*; *Paoloni et al., 2009*). For GSE36001, we obtained 19 osteosarcoma cell lines and six normal samples, including two normal osteoblast samples and four normal bone samples. For GSE39055 and GSE16091, we obtained 37 and 34 human osteosarcoma samples, respectively.

### EMT score

The EMT-related gene signature comprised 200 genes obtained from the gene set "HALLMARK_EPITHELIAL_MESENCHYMAL_TRANSITION" in The Molecular

Signatures Database hallmark gene sets (MSigDB, software.broadinstitute.org/gsea/msigdb, Table S2) (*Subramanian et al., 2005*). For each sample in The Cancer Genome Atlas (TCGA), the EMT score was calculated by ssGSEA in R software 3.5.0 based on EMT-related gene expression (*Hanzelmann, Castelo & Guinney, 2013*). ssGSEA calculates the EMT gene set enrichment score per osteosarcoma sample as the normalized difference in empirical cumulative distribution functions of EMT-related gene expression ranks inside and outside the EMT gene set.

### Immune score, stromal score and tumor purity
To explore the relationship among EMT, immune activity, stromal activity and tumor purity, we used the ESTIMATE package of R software 3.5.0 to evaluate the index (*Yoshihara et al., 2013*). The ESTIMATE package defines immune and stromal gene signatures to infer the proportion of the immune and stromal components from gene expression data and combines these individual components to estimate tumor purity. The stromal gene signature comprises 141 genes.

### Correlation analysis
Correlation analysis was used for comparisons of two gene sets using GraphPad Prism 7. The Pearson $R$ value and two-tailed $P$-value are used to evaluate the correlation level. Compute the $R$ value for $X$ vs every $Y$ data set.

### Expression analysis of EMT and stromal cell gene signature
GSE36001 was used in the expression analysis of EMT-related and stromal cell gene signatures. The expression value was vst-transformed and quantile normalized. Fold change (FC) in EMT-related and stromal cell genes between tumor and normal samples was calculated. Student's $t$-test was used for comparisons between two groups using GraphPad Prism 7.

### Survival analysis
The overall survival (OS) and recurrence-free survival (RFS) data from GSE39055 was used for survival analysis. Kaplan–Meier survival curves and the log-rank test were used to assess the OS and RFS and index score or individual gene expression. The degree of OS and RFS were summarized by hazard ratios (HRs) and $P$-values, which were determined using GraphPad Prism 7. The method of determining cut-off values for grouping high and low levels was described in previous studies (*Uhlen et al., 2017*).

### Real-time quantitative polymerase chain reaction
Five osteosarcoma tissues and their pair-matched adjacent stromal tissues were obtained with informed consent from patients who underwent radical resections at the Department of Orthopedics, The Third People's Hospital of Chengdu, China. Osteosarcoma and stromal tissues were derived from surgical resection of osteosarcoma. The expression levels of 25 EMT-related genes in the osteosarcoma and stromal samples were examined using RT-qPCR. RNA was converted into cDNA using the PrimeScript® 1st Strand cDNA Synthesis Kit (TaKaRa, Kusatsu, Japan). Quantification of the cDNA template was

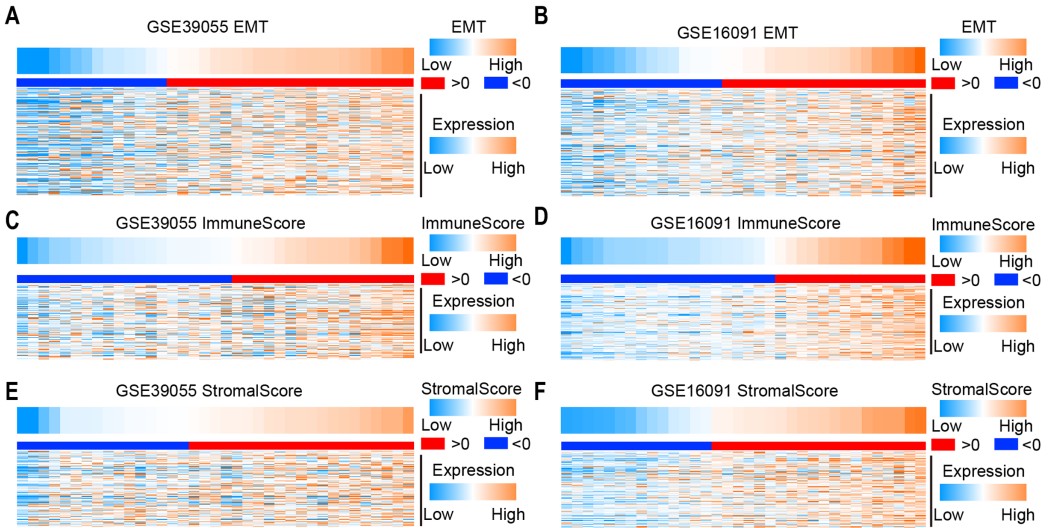

**Figure 1 Expression of EMT-, immune activity and stromal-related genes in osteosarcoma.** (A–F) Heatmap showing the expression of EMT (A and B), immune activity (C and D) and stromal-related (E and F) genes in osteosarcoma in two independent microarray datasets (GSE39055 on the left and GSE16091 on the right).

performed with RT-qPCR using SYBR green as a fluorophore. The primer sequences of the 25 EMT-related genes examined in this study can be found in the Table S1. All experimental procedures were approved by the Medical Ethics Committee of the Third People's Hospital of Chengdu.

# RESULTS

## EMT-related genes, immune activity and stromal genes in osteosarcoma

To determine the EMT score in osteosarcoma patients, we collected EMT-related genes from the gene set "Hallmark_epithelial_mesenchymal_transition" in MSigDB. The EMT score was calculated by ssGSEA according to gene expression in two independent microarray datasets (GSE39055 and GSE16091). Obviously, the expression level of EMT-related genes increased as the EMT score increased in GSE39055 and GSE16091 (Figs. 1A and 1B). To identify immune and stromal activity in osteosarcoma patients, the immune score and stromal score were calculated by ESTIMATE according to gene expression in GSE39055 and GSE16091. We found that the expression of immune-related genes and stromal related genes increased as the immune score (Figs. 1C and 1D) and stromal score (Figs. 1E and 1F) increased, respectively. These results suggest that the EMT, immune and stromal scores can be used to evaluate the expression of genes related to EMT, immune activity and stromal activity, respectively.

## EMT-related gene expression is positively correlated with immune and stromal activity in osteosarcoma

We explored the relationship among EMT, immune activity and stromal activity. The EMT score was positively correlated with the immune score in GSE39055

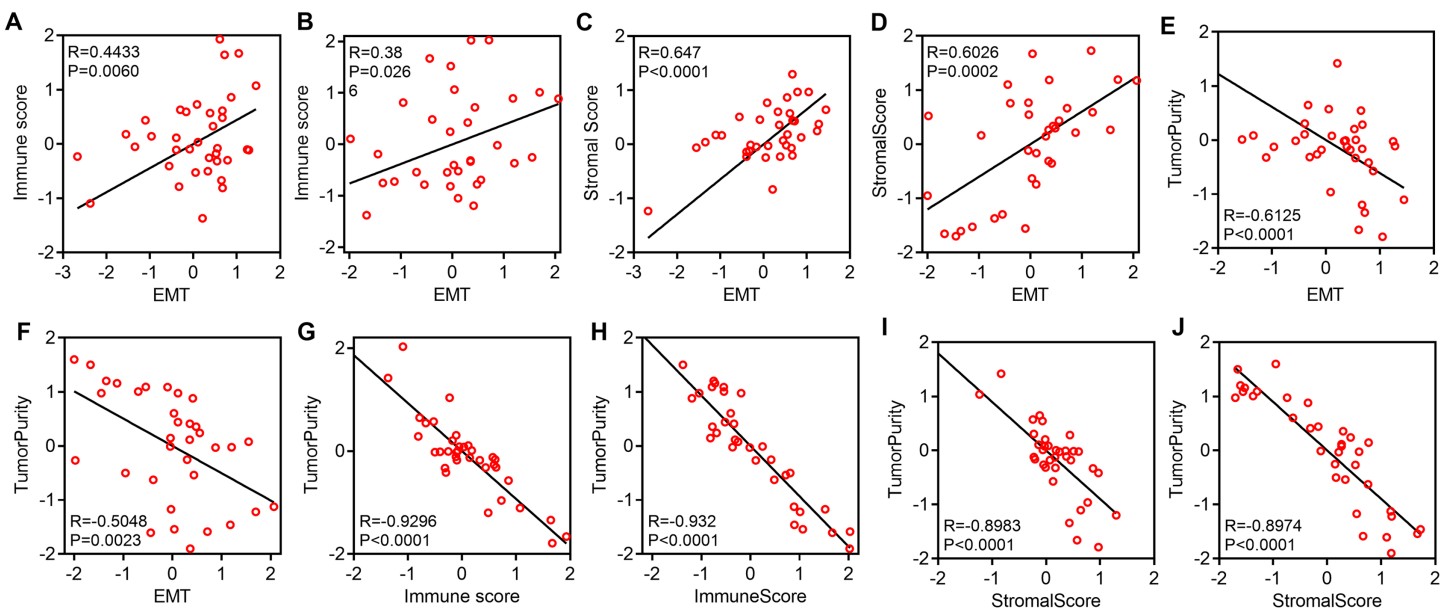

**Figure 2** **EMT-related gene expression is positively correlated with immunity and stromal activity but negatively correlated with tumor purity.**
Plot showing the correlation between EMT and immunity in GSE39055 (A) and GSE16091 (B). Plot showing the correlation between EMT and stromal activity in GSE39055 (C) and GSE16091 (D). Plot showing the correlation between EMT and tumor purity in GSE39055 (E) and GSE16091 (F). Plot showing the correlation between immunity and tumor purity in GSE39055 (G) and GSE16091 (H). Plot showing the correlation between stromal activity and tumor purity in GSE39055 (I) and GSE16091 (J).                 

($R = 0.4433$, $P = 0.0060$) and GSE16091 ($R = 0.38$, $P = 0.0266$) (Figs. 2A and 2B). The EMT score was also positively correlated with the stromal scores in GSE39055 ($R = 0.647$, $P < 0.0001$) and GSE16091 ($R = 0.6026$, $P = 0.0002$) (Figs. 2C and 2D). Next, we analyzed the relationship among tumor purity, EMT, immune activity and stromal activity. We found that the EMT score was negatively correlated with tumor purity in GSE39055 ($R = −0.6125$, $P < 0.0001$) and GSE16091 ($R = −0.5048$, $P = 0.0023$) (Figs. 2E and 2F). The immune score was negatively correlated with tumor purity in GSE39055 ($R = −0.9296$, $P < 0.0001$) and GSE16091 ($R = −0.932$, $P < 0.0001$) (Figs. 2G and 2H). The stromal score was also negatively correlated with tumor purity in GSE39055 ($R = −0.8983$, $P < 0.0001$) and GSE16091 ($R = −0.8974$, $P < 0.0001$) (Figs. 2I and 2J). These results indicate that EMT-related gene expression in osteosarcoma may result from stromal cells in the tumor micro environment rather than epithelial cancer cells.

## Expression of EMT-related genes and immune activity result in disparate clinical outcomes in osteosarcoma patients

To explore the clinical value of EMT and immune activity in osteosarcoma, we analyzed OS and RFS in GSE39055. OS is an important endpoint, with the advantage of minimal ambiguity in defining an OS event (*Liu et al., 2018*), while RFS can represent disease recurrence under the influence of related factors. Figures 3A–3D demonstrates the association of OS with EMT and immune activity in osteosarcoma patients. A high EMT activity was associated with significantly poorer OS in osteosarcoma patients ($HR = 4.977$, $P = 0.0025$, Fig. 3A). However, a high immune score was associated with

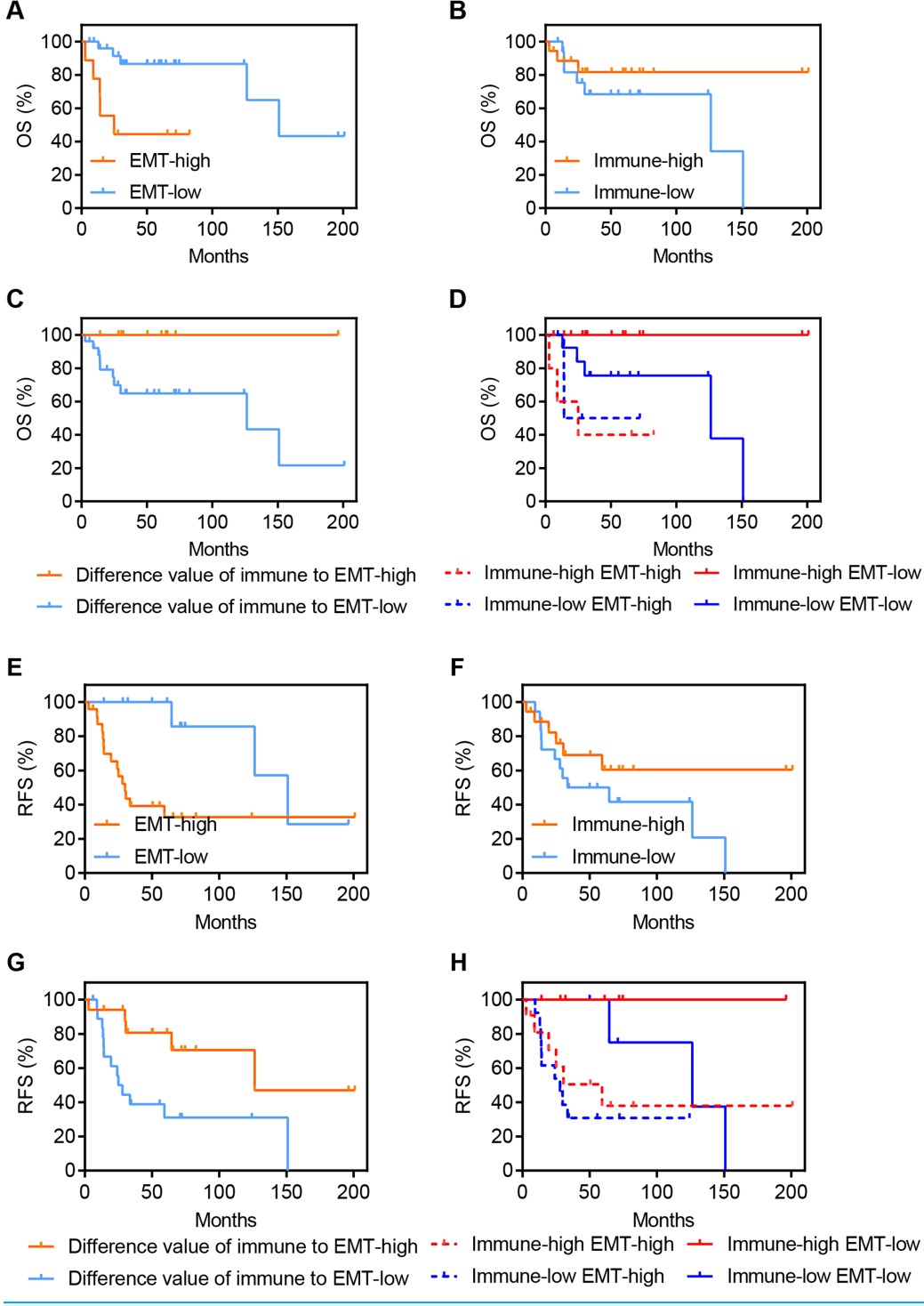

**Figure 3 Expression of EMT-related genes and immune activity result in disparate clinical outcomes in osteosarcoma patients.** (A–D) Kaplan–Meier survival curves showing the OS of osteosarcoma patients in GSE39055 divided into two groups by the (A) EMT score, (B) immune score and (C) difference between the immune and EMT score or (D) divided into four groups by both EMT and immune score. (E–H) Similar to A–D, but demonstrate the RFS of osteosarcoma patients in GSE39055.

improved OS in osteosarcoma patients, but this association was not significant (HR = 0.4047, $P$ = 0.1726, Fig. 3B). Although the EMT score was positively correlated with immune activity (Figs. 2A and 2B), disparate clinical outcomes were reported. Thus, we explored whether combining these factors would have further prognostic value in these patients. We found that a large difference between immune activity and the EMT score was associated with significantly improved OS in osteosarcoma patients (HR = 0, $P$ = 0.029, Fig. 3C). Figure 3D shows the OS of four groups stratified according to both EMT and immune scores ($P$ = 0.007). High immune activity and a low EMT score were associated with the best OS in osteosarcoma patients. However, low immune activity and a high EMT score were associated with significantly poorer OS in osteosarcoma patients.

Figures 3E–3G presents the association of RFS with EMT and immune activity in osteosarcoma. Similar to the results of OS in patients, a high EMT score was associated with significantly poorer RFS in osteosarcoma patients (HR = 4.308, $P$ = 0.0098, Fig. 3E). A high immune score was associated with better RFS in osteosarcoma patients, but this association was not significant (HR = 0.4816, $P$ = 0.1335, Fig. 3F). A large difference between the immune activity and EMT scores was associated with significantly improved RFS in osteosarcoma patients (HR = 0.2781, $P$ = 0.0085, Fig. 3G). High immune activity and low EMT were associated with the best RFS in osteosarcoma patients, while low immune activity and high EMT were associated with the worst RFS (Fig. 3H).

## Stromal cells are a key source of EMT-related gene expression in osteosarcoma patients

To explore the relationship between EMT-related and stromal cell gene signatures in osteosarcoma, we analyzed the expression of and clinical outcomes associated with each gene signature. First, we analyzed related gene expression in the GSE36001 dataset. At the individual gene level, the expression of EMT-related genes was higher than that of stromal cell genes; the mean value of EMT-related genes was 9.12, while that of stromal cell genes was 7.421 ($P$ < 0.0001, Figs. 4A and 4B). The FC in EMT-related genes was also higher than that in stromal cell genes; the mean value of EMT-related genes was −0.98, while that of stromal cell genes was −0.7 ($P$ < 0.0001, Figs. 4C and 4D). Next, we evaluated the prognostic significance of individual genes from EMT and stromal signatures. Ten of the top 15 genes and 11 of the top 15 genes most significantly associated with OS and RFS in GSE39055 were from the EMT-related gene set (Figs. 4E and 4F).

To verify our findings, we used RT-qPCR to detect the relative expression of the above 25 genes, including 18 EMT signature genes, three stromal signature genes and four genes belonging to both EMT-related and stromal signatures. We found that the expression of EMT-related genes was higher than that of stromal cell genes in human osteosarcoma samples (Fig. 4G), while the expression of stromal cell genes was higher than that of EMT-related genes in human stromal samples (Fig. 4H). Moreover, the FC in EMT-related genes was also higher than that in stromal cell genes (Fig. 4I). Therefore, although EMT and stromal activity were highly correlated in osteosarcoma (Figs. 2C and 2D), EMT-related genes were more strongly associated with survival than stromal cell genes.

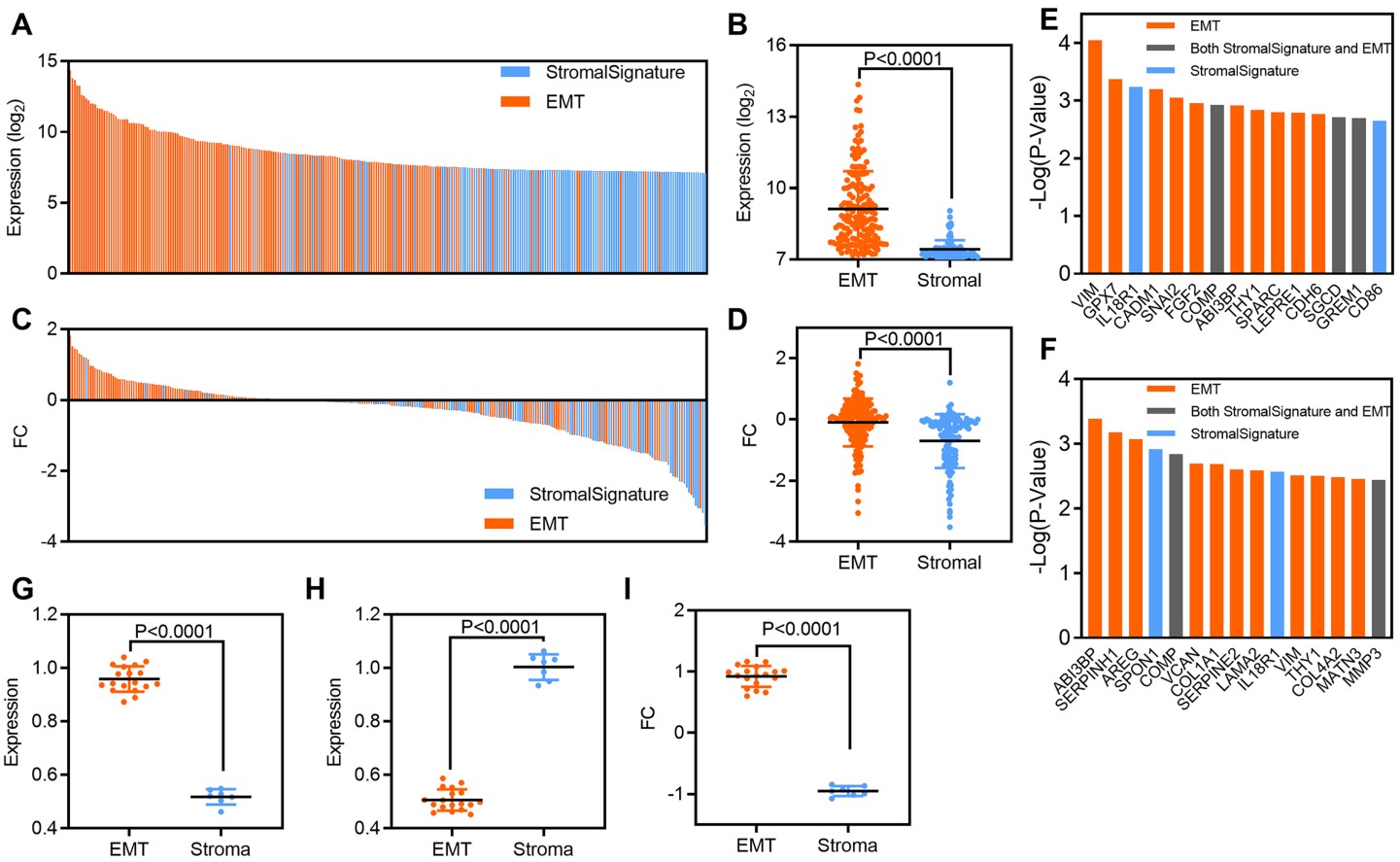

**Figure 4 EMT-related gene expression originates predominantly from stromal cells in osteosarcoma patients.** (A) Bar plot showing the expression of EMT-related and stromal gene signatures in GSE36001 osteosarcoma samples. (B) Scatter plot showing expression grouped by EMT-related and stromal gene signatures in GSE36001 osteosarcoma samples. (C) Bar plot showing the FC in EMT-related and stromal gene signatures in GSE36001. (D) Scatter plot showing the FC in EMT-related and stromal gene signatures in GSE36001. (E and F) Bar plot showing individual EMT-related and stromal genes ranked according to the significance of their association with OS and RFS in GSE39055. (G–I) Scatter plot showing expression of grouped by EMT-related and stromal (including both in stromal and EMT) gene signatures in osteosarcoma (G), stromal samples (H) and FC (I) by using RT-qPCR.           

These results suggest that stromal cells may serve as a key source of EMT-related gene expression in osteosarcoma.

## DISCUSSION

Osteosarcoma is the most common type of bone cancer in young people. The 5 year survival rate for patients who have metastatic disease is low (*Chang, 2019*; *Maximov et al., 2019*; *Shen et al., 2017*). Tumor metastasis begins with EMT, which is a critical event in osteosarcoma metastasis (*Jiang et al., 2019*; *Shen et al., 2017*; *Wang et al., 2018*). Moreover, accumulating evidence has shown that EMT is associated with immunity in human cancers (*Lou et al., 2016*; *Mak et al., 2016*; *Singh & Chakrabarti, 2019*). However, to the best of our knowledge, the relationship between EMT and the immune micro environment in osteosarcoma has not been reported. Here, we have shown that EMT was positively correlated with immune activity and that these parameters may hold prognostic value for osteosarcoma patients.

We defined EMT-related gene signatures and evaluated immune activity and stromal activity based on gene expression in two independent micro array datasets. EMT and immune activity demonstrated a positive correlation. A positive correlation was also observed between EMT and stromal cell activity. Furthermore, we found that tumor purity was negatively correlated with EMT, immune activity and stromal cells. Altogether, these results suggest that the expression of EMT-related genes in osteosarcoma may originate from stromal cells in the tumor microenvironment rather than from epithelial cancer cells. These findings are consistent with the findings from recent studies in urothelial cancer and colorectal cancer (*Isella et al., 2015*, *2016*; *Wang et al., 2018*). Using TCGA urothelial cancer dataset, *Wang et al. (2018)* showed that gene expression-based measures of infiltrating T-cell abundance and EMT are positively correlated. Stromal cells are a major source of EMT-related gene expression in bulk urothelial cancer transcriptomes. Using colorectal cancer expression data from patient-derived xenografts, *Isella et al. (2015)* showed that upregulated EMT-related genes are also prominently expressed by stromal cells, suggesting that EMT-related genes could originate from stromal rather than epithelial cancer cells.

We demonstrated that the expression of EMT-related genes and immune activity result in disparate clinical outcomes in osteosarcoma patients. A high EMT score was associated with significantly poor OS and RFS in osteosarcoma patients, while a high immune score was associated with better OS and RFS in osteosarcoma patients, but the latter association was not significant. However, combining these factors had further prognostic value for osteosarcoma patients in terms of OS and RFS. We found that high immune activity and a low EMT score were associated with the best OS and RFS in osteosarcoma patients. Nevertheless, low immune activity and a high EMT score were associated with significantly poorer OS and RFS in osteosarcoma patients. These findings clarify the predictive of EMT-related gene expression and immune activity for survival and recurrence in osteosarcoma patients.

To further explore the source of EMT-related gene expression, we analyzed the gene expression in microarray datasets from the GEO and human samples that we collected. The expression of EMT-related genes was higher than that of stromal cell genes in the osteosarcoma samples, while the expression of stromal cell genes was higher than that of EMT-related genes in the human stromal samples. The FC of EMT-related genes between tumor and normal (stromal) samples was higher than that in stromal cell genes. Although EMT and stromal activity were highly correlated in osteosarcoma, EMT-related genes were more strongly associated with survival than stromal cell genes. These results suggested that stromal cells may serve as a key source of EMT-related gene expression in osteosarcoma. Taken together, we show that EMT gene expression and immunity are positively correlated in osteosarcoma yet confer a disparate treatment response and prognostic information. EMT-related gene expression in osteosarcoma may require reinterpretation, given the key contribution of stromal cells to such gene expression. The balance of EMT/stromal vs immunity elements may provide more information about the antitumor immune response than measures of immunity alone.

## CONCLUSIONS

Taken together, the results of our study clarified the relationship among EMT, immune activity, stromal activity and tumor purity in osteosarcoma through an analysis of gene expression and evaluation of clinical prognosis. Future research could employ single cell RNA-seq technology to further study these factors in osteosarcoma.

### Funding
The authors received no funding for this work.

### Competing Interests
The authors declare that they have no competing interests.

### Author Contributions
- Yin-xiao Peng conceived and designed the experiments, performed the experiments, analyzed the data, prepared figures and/or tables, authored or reviewed drafts of the paper, and approved the final draft.
- Bin Yu conceived and designed the experiments, performed the experiments, analyzed the data, prepared figures and/or tables, authored or reviewed drafts of the paper, and approved the final draft.
- Hui Qin performed the experiments, prepared figures and/or tables, and approved the final draft.
- Li Xue conceived and designed the experiments, authored or reviewed drafts of the paper, and approved the final draft.
- Yi-jian Liang conceived and designed the experiments, authored or reviewed drafts of the paper, and approved the final draft.
- Zheng-xue Quan analyzed the data, authored or reviewed drafts of the paper, and approved the final draft.

### Human Ethics
The following information was supplied relating to ethical approvals (i.e., approving body and any reference numbers):

Medical Ethics Committee of the Third People's Hospital of Chengdu granted Ethical approval to carry out the study within its facilities.

### Data Availability
The raw measurements are available in the Supplemental Files.

### Supplemental Information
Supplemental information for this article can be found online at http://dx.doi.org/10.7717/peerj.8489#supplemental-information.

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
