# Peer review of "EMT-related gene expression is positively correlated with immunity and may be derived from stromal cells in osteosarcoma"

_PeerJ, doi:10.7717/peerj.8489_

## Round 0.1 · original submission · Major Revisions

Three reviewers and myself have read your manuscript and our opinion is that more cautious reporting and interpretation should be carefully added. Please take into account reviewers minor and major editing requests below.

·

Basic reporting

This is an interesting manuscript that shows the relationship of EMT genes, immune activity, stromal and tumor purity in osteosarcoma. EMT-related gene is positively correlated with immune and stromal activity in osteosarcoma. High immune activity and low EMT were associated with the best OS and RFS in osteosarcoma patients. Although EMT and stromal activity were highly correlated in osteosarcoma, EMT-related genes were more strongly associated with survival than stromal cell genes. The authors claimed stromal cells as a key source of EMT-related gene expression in osteosarcoma patients.

Experimental design

The bioinformatics analysis based on the collected patients data and public datasets are reasonable. The experiment also verified the results of the bioinformatics results.

Validity of the findings

In result 2, why the author claimed the expression of EMT-related gene may result from stromal cells rather than epithelial cancer cells activity in the tumor microenvironment?

Additional comments

This paper provides interesting data confirming the relationship of EMT genes, immune activity, stromal and tumor purity in osteosarcoma.
The following are some minor weakness:
1. The abbreviation “RT-PCR” should be “RT-qPCR”, such as line 31 and 75.
2. The author used 200 EMT related genes obtained from the MSigDB. These genes should be listed in supplementary materials.
3. Inconsistent format P-value, for example, P<0.0001 in line 139, but p<0.0001 in line 143.

Reviewer 2 ·

Basic reporting

In this manuscript, the study investigated the relationship between EMT-related genes expression, mutation and immune response in osteosarcoma. The article is interesting and compatible with the area of interests of the journal. The experiments are well organized and clear.

Experimental design

The data in this study was derived from three independent microarray datasets GSE36001, GSE39055 and GSE16091. What are the criteria for selecting them?

Validity of the findings

None.

Additional comments

1. Abbreviations and acronyms are typically defined the first time the term is used within the main text and then used throughout the remainder of the manuscript. Please consider adhering to this convention. such as, OS and RFS in abstract (line 34).
2. Please briefly describe the calculation method of EMT score in the methods (line 86-90), so as not to cause confusion to readers.
3. In legend Fig.4A, bar plot showing the expression of EMT-related and stromal gene signatures in GSE36001 osteosarcoma samples. However, Y-axis show FC. Also, the same in Fig.4C. Did the author reverse the order of 4A and 4C? please verify.
4. The authors find that the expression of EMT genes and immunity are positively correlated, these signatures convey disparate prognostic information. Furthermore, EMT gene expression could derive from stromal rather than epithelial cancer cells. What do these findings specifically reflect in clinical value? What is the innovation of this research? I hope the author can reflect this in the discussion.

·

Basic reporting

No Comment.

Experimental design

No Comment.

Validity of the findings

No Comment.

Additional comments

Peng et al. show that there is a strong gene expression correlations among genes related to EMT, immunity, and stromal profile in the context of osteosarcoma using publicly available datasets. Interestingly, these gene expressions are negatively correlated with tumor purity. The authors also stratified patient survival data and found that EMT is more predictive of poor survival (OS, RFS) than immune-related genes and that combining their signatures can provide better prognostic value. They also suggest that EMT gene expression is generally derived from stromal cells rather than epithelial cancer cells.

While the manuscript contain commendable elements (i.e., providing insights of gene activity in osteosarcoma), the current study heavily relied on gene expression profiles and patient survival correlations, which may or may not be directly coupled physiologically. I find the current manuscript lacking of sufficient experimental bolstering. Regardless, the authors can still improve the manuscript. There are several issues I think the authors should address in this regard:

1. Proper descriptions on the controls (non-cancer) used should be evident in the manuscript, especially in the methods section.
2. The manuscript also lack the discussion of mechanistic implications of their findings. Reflection on mechanisms should at least be discussed in several parts of the results and discussion sections. However, I should note that this be done cautiously as the authors did not couple their correlations with in vitro or in vivo experiments (which is the main lacking element of the paper).
3. Conclusions on the EMT lineage formation are presumptuous if the authors are only basing their results on 1 gene expression dataset and patient survival correlations. They also used a very small number of stromal gene markers. More significant stromal markers should be included in this analysis in addition to more cautious reporting of the EMT gene expression lineage formation.
4. The methods section is not descriptive enough. Please provide more description on how each association/correlation analysis was done.
There are many grammar lapses all throughout the manuscript from the title to the figure legends. I recommend a complete review of grammar, paragraph structures, and paraphrasing knowledge referenced from various sources.

---

## Round 0.2 · accepted · Accept

You have addressed in a satisfactory manner the concerns of the three reviewers.

·

Basic reporting

No comment.

Experimental design

No comment.

Validity of the findings

No comment.

Additional comments

The authors answered all my concerns and the article is acceptable.

Reviewer 2 ·

Basic reporting

No comment

Experimental design

No comment

Validity of the findings

No comment

Additional comments

No comment

·

Basic reporting

No comment

Experimental design

No comment

Validity of the findings

No comment

Additional comments

The authors have substantially revised their manuscript, which enhanced the clarity of the
manuscript. I have no further comments to the authors.